



# RadNet 1.0: Exploring deep learning architectures for longwave radiative transfer

Ying Liu[1], Rodrigo Caballero[1], and Joy Merwin Monteiro[1]

[1]Department of Meteorology, Stockholm University, Stockholm, Sweden

**Correspondence:** Ying Liu (ying.liu.sweden@gmail.com)

**Abstract.** Simulating global and regional climate at high resolution is essential to study the effects of climate change and capture extreme events affecting human populations. To achieve this goal, the scalability of climate models and efficiency of individual model components are both important. Radiative transfer is among the most computationally expensive components in a typical climate model. Here we attempt to model this component using a neural network. We aim to study the feasibility of replacing an explicit, physics-based computation of longwave radiative transfer by a neural network emulator, and assessing the resultant performance gains. We compare multiple neural-network architectures, including a convolutional neural network and our results suggest that the performance loss from the use of convolutional networks is not offset by gains in accuracy. We train the networks with and without noise added to the input profiles and find that adding noise improves the ability of the networks to generalise beyond the training set. Prediction of radiative heating rates using our neural network models achieve up to 370x speedup on a GTX 1080 GPU setup and 11x speedup on a Xeon CPU setup compared to the a state of the art radiative transfer library running on the same Xeon CPU. Furthermore, our neural network models yield less than 0.1 Kelvin per day mean squared error across all pressure levels. Upon introducing this component into a single column model, we find that the time evolution of the temperature and humidity profiles are physically reasonable, though the model is conservative in its prediction of heating rates in regions where the optical depth changes quickly. Differences exist in the equilibrium climate simulated when using the neural network, which are attributed to small systematic errors that accumulate over time. Thus, we find that the accuracy of the neural network in the "offline" mode does not reflect its performance when coupled with other components.

## 1   Introduction

Computational models of Earth's climate are essential tools to advance our understanding of the climate system and our ability to predict its response to perturbations such as increased levels of greenhouse gases. Climate models contain algorithmic representations of the various components of the climate system like the atmosphere, ocean, sea ice and land surface. Our ability to predict future changes in climate depends crucially on the accuracy of these models and the extent to which interactions between various components of the climate system are represented.

A basic requirement for increased model fidelity, particularly at the regional scale, is increased spatial resolution. However, the computational burden increases roughly as the fourth power of spatial resolution (since resolution must increase along all



three spatial dimensions, and the time step reduced to ensure numerical stability). To address this problem, various approaches have been used including improved model scalability (Dennis and Loft, 2011) and the use of low-precision floating point operations (Palmer, 2014).

Long simulations using high resolution climate models are needed to explore key questions in climate research, particu-
larly changes in the statistics weather extremes such as windstorms and precipitation events. Radiative transfer (RT) in the atmosphere is among the most computationally burdensome components of such simulations. While the basic equations for calculating RT are straightforward, the complex nature of the absorption bands of greenhouse gases such as carbon dioxide and water vapour requires separate calculation over a very large number of small spectral intervals to obtain accurate results. Since such a line-by-line calculation is extremely computationally intensive and not feasible in a realistic climate model integration, it
is necessary to group individual absorption lines into bands or clusters with similar properties as in the correlated-k method (Fu and Liou, 1992). Such methods can dramatically improve the computational performance while retaining adequate accuracy in the computation. Many state-of-the-art climate models use the Rapid Radiative Transfer Model for General circulation models (RRTMG). RRTMG is based on the single-column correlated k-distribution reference model RRTM (Iacono et al., 2008b). RRTMG tries to strike a balance betweeen computational complexity and accuracy by reducing the number of calculations per
band while ensuring fidelity with the RRTM code (Iacono et al., 2008a). Nonetheless, even when employing such simplified schemes, RT remains amongst the most numerically expensive components of climate models, and a variety of strategies have been developed to reduce this cost (see for example Pincus and Stevens, 2013, and references therein).

In this paper, we explore the potential performance gains achievable by using a neural network (NN) to calculate radiative transfer. Specifically, we train a variety of alternative NN architectures on a set of radiative heating rate profiles computed
using a state-of-the-art RT code (see Sections 2), and compare the computational performance of the NN with that of the RT code itself. Note that this comparison only serves to assess the performance of RT calculation in standalone form. We expect a suitably-trained neural network to be a drop-in replacement for the RT code in a full climate model, and expect that other computational costs—such as data transfer within and between computational nodes—will not change, but we do not explicitly address this issue in this exploratory study. Instead, our focus here is on identifying the most suitable NN architecture in
terms of accuracy and computational performance. We also explore the behaviour of the NN in a time-evolving single-column radiative-convective model (Section 4).

Recent advances in NNs have led to rapid progress in the accuracy of pattern and image recognition tasks. In particu-
lar, convolutional neural networks (CNNs) (Krizhevsky et al., 2012a) have achieved impressive results for image classifica-
tion (Krizhevsky et al., 2012b), while recurrent neural networks (RNNs) have made breakthroughs in sequence-to-sequence
learning tasks such as machine-translation (Wu et al., 2016). Efforts to use machine learning techniques to model actual phys-
ical processes in a climate model have increased recently (Schneider et al., 2017; Gentine et al., 2018; Rasp et al., 2018;
O'Gorman and Dwyer, 2018; Scher, 2018; Brenowitz and Bretherton, 2018; San and Maulik, 2018). In particular, it is now being recognized that physical processes whose representation in climate models has usually been inexact and parameterised could potentially be improved by using machine learning techniques. RT, on the other hand, has always been an attractive
candidate to optimize in climate models because of the large computational cost, as discussed above. Optimization has been



attempted using traditional optimization, porting to new architectures such as GPUs (Price et al., 2014; Mielikainen et al., 2016; Malik et al., 2017) and using NNs to approximate RT. Initial attempts to retrieve radiative heating profiles used shallow (one hidden layer) networks (Chevallier et al., 1998), and similar NN architectures were successfully used to replace RT in decadal simulations using conventional climate models (Krasnopolsky et al., 2005, 2008, 2009). Recently, a deep NN was used
to replace RT in a high resolution GCM, and was successfully used to run the GCM for one year (Pal et al., 2019). These studies show the capability of NNs to accurately approximate radiative heating profiles in a particular climate regime, while raising questions about how generalizable this learning actually is in terms of handling perturbed climate states. Studying the effect of perturbations (in sea-surface temperature, greenhouse gases, aerosols or cloud properties) on the climate of a model is a very typical use-case in climate science, and the performance of NNs in such scenarios has yet to be studied carefully.
In the context of machine learning for climate modelling applications, the following questions are still not well understood (Dueben and Bauer, 2018):

– What NN architectures are most suitable?

– What is the accuracy-efficiency tradeoff between different NN architectures?

– What accuracy loss can we expect when the NN is provided with "non-typical" input values, i.e. values very different
from those in the training sample, such as would occur in a perturbed climate experiment?

– What is the speed-up we can expect by replacing a traditional RT scheme with a NN?

Our aim here is to address these four questions. To limit the scope of this exploratory study, we focus on longwave radiative transfer under clear-sky conditions (henceforth, RT thus refers to clear-sky longwave radiative transfer). We use the RRTMG library available within the climt modelling toolkit (Monteiro et al., 2018) to generate radiative cooling profiles to train the
NN models. In particular, we compare the accuracy-computational complexity tradeoff between five kinds of NN architectures on both CPU and GPU. We also study the loss in accuracy if perturbations are added to the input. The question of accuracy loss is all the more relevant in RT due to its mathematical structure – since RT is modelled as an integral equation, localised perturbations have global impacts on the profile of radiative heating or cooling obtained.

The paper is organized as follows. The preparation of data for training and validation of the NNs is presented in Section 2.
Section 3 presents the NN structures and parameters we have used. Evaluation results are presented in section 4. Finally, we present a brief discussion along with concluding remarks in Section 5.

## 2   Data and Methods

While radiative transfer is inherently three dimensional, increasing its complexity and computationally cost, it is common to assume horizontal homogeneity (independent column assumption) and retain only a single (vertical) dimension (Meador
and Weaver, 1980). This independent colummn assumption underlies almost all raditive transfer codes used in weather and climate models, and reduces radiative transfer calculation to an "embarassingly parallel" 1-dimensional problem in each vertical





column of the atmosphere. For a given longitude-latitude point, RT can be represented by a vector whose length is the number of vertical levels into which the column is discretized. The calculation of RT under clear sky (cloud-free) conditions is based on a number of inputs, including vectors of atmospheric pressure, air temperature and specific humidity at each level, while
surface temperature and carbon dioxide mixing ratio are represented as scalars. While the clear-sky RT in the atmosphere is affected by other greenhouse gases like methane and aerosols like sulphates, we restrict ourselves to using the above quantities in this exploratory study.

## 2.1   The ERA-Interim Dataset

We use the ERA-Interim dataset (Dee et al., 2011) to provide temperature and humidity profiles for training the neural network.
We use 6 hourly model-level data, which has a higher resolution in the vertical as compared with the pressure level data. This implies that pressure is not a constant and is therefore an additional input to the neural network.

The ERA-Interim dataset consists of 38 years of data spanning the period 1979 to 2016, which amounts to around 6.5 billion sample profiles. We employ the first 7 years of ERA-Interim historical data as the training dataset, i.e., data from 1979 to 1985 and the last 2 years of the ERA-Interim historical data as the validation dataset, i.e., data from 2015 to 2016. Considering the
model training time, we have applied random sampling of 1% with respect to each year in the training and validation datasets. After sampling, we name the training dataset as $Dataset_1$ and the validation dataset as $Dataset_{1.val}$. The reason for using this data separation schema is because that we would like to examine whether our radiation prediction model is able to generalize to unseen/future data inputs while being trained on the oldest data.

## 2.2   Perturbed Dataset

Figure 1 shows the mean and variance of ERA-Interim air temperature, humidity and radiative heating rates calculated using RRTMG from 1979 to 2016. Using the above statistics, we have augmented our training data by created a perturbed dataset as follows:

1. Pick an original profile from the historical samples.

2. Generate a random air temperature profile assuming Gaussian distribution at each vertical level using the statistics from
Figure 1.

3. Generate a random weight (between -0.2 to 0.2) for the generated air temperature profile.

4. Generate an augmented air temperature profile by adding together the original profile with the weighted random profile vertical level wise.

5. Calculate the maximum humidity given the air temperature and pressure at each vertical level.

6. Calculate the original relative humidity ratio using humidity divided by the maximum humidity at each vertical level.

7. Calculate the new maximum humidity given the generated air temperature and pressure at each vertical level.



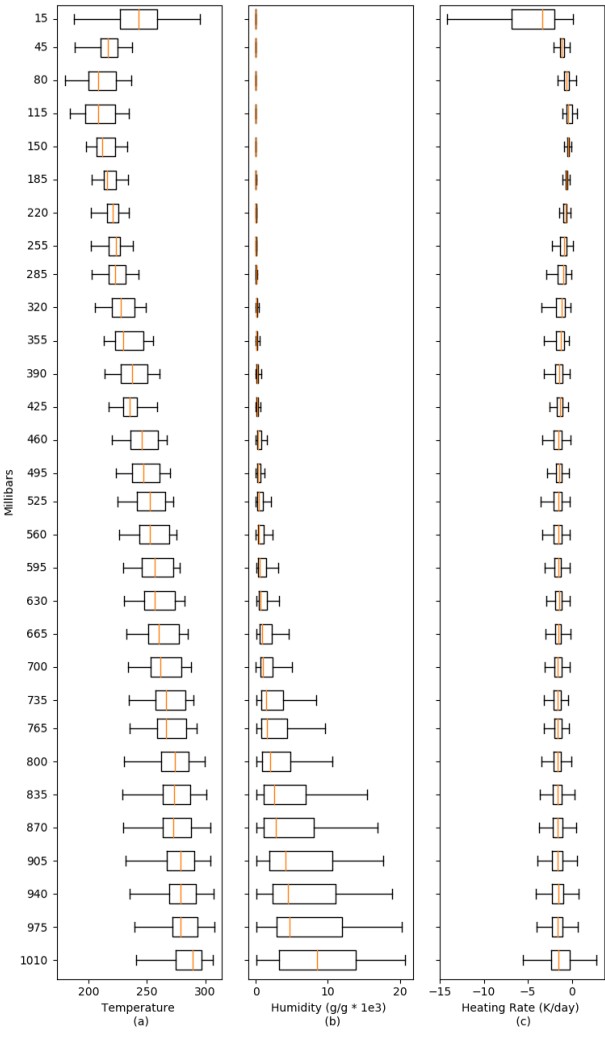

**Figure 1.** (a,b) ERA-Interim air temperature (K) and humidity (g kg$^{-1}$) statistics. (c) Longwave radiative rates (K day$^{-1}$) calculated using RRTMG. Vertical axis is pressure in Pa.

8. Generate the corresponding humidity by multiplying the new maximum humidity and the original relative humidity ratio at each vertical level.

9. We keep the surface temperature and the carbon dioxide mixing ratio the same as the original profile.

Augmented datasets are generated using $Dataset_1$ and $Dataset_{1.val}$. Then, the augmented dataset are 50-50 mixed with $Dataset_1$ and $Dataset_{1.val}$ respectively to create $Dataset_2$ and $Dataset_{2.val}$. The purpose of generating $Dataset_2$ and $Dataset_{2.val}$ is that we would like to use it to investigate the generality of our RT prediction model. The specific evaluation procedures are described in the evaluation section.





## 2.3   The RT dataset

The calculation of the radiative fields for the historical and perturbed datasets are calculated using the RRTMG component available in the climt modelling toolkit (Monteiro and Caballero, 2016; Monteiro et al., 2018). This component is a python wrapper over the RRTMG fortran library, and provides convenient access to the radiation fields. The statistics of the generated radiative heating profiles are also shown in Figure 1.

## 3   Neural Network Models

### 3.1   Neural Network Basics

A neural network is composed of multiple neurons, or even multiple layers of neurons in order to model complex scenarios. A simple neural network is a feed-forward network where information flows only in one direction from input to output. Multilayer perceptron (Gardner and Dorling, 1998) is the most common feed-forward NN. It consists of an input layer that passes the input vector to the network, one or more hidden layers and an output layer. There are usually activation functions applied in each
layer. An activation function usually introduces non-linearity in order to allow a NN to tackle with complicated problems and learn complex representations.

Convolutional NN is another type of NN designed for image-focused tasks. It is widely used in many fields such as image classification, object detection and image segmentation (Krizhevsky et al., 2017). CNNs usually consist of three types of layers, convolutional layers, pooling layers and fully-connected layers. A convolutional layer is composed of learnable kernels
or filters. The kernel usually considers a small region of input at one time, but covers the entirety of the input. Specifically, it slides over the input spatially and computes dot products between the kernel and the area of input covered by the kernel. With each kernel, a convolutional layer produces an activation map, whose size depends on whether there is a stride or padding. All the activation maps will be stacked together along the depth dimension and passed on to the next layer (O'Shea and Nash, 2015). Neurons in a layer are connected to only a small region of the previous layer instead of everything, which is different
from feedforward neural networks. In this way, convolutional layers are better at extracting locality-dependent features, such as shapes and patterns in images.

In the context of RT, we use CNNs to evaluate whether the sensitivity to localised features improves the prediction performance of deep neural networks. In particular, strong local changes in the optical properties of the atmosphere are fairly common due to the presence of clouds or horizontal advection of water vapour. While this work focuses on clear-sky radiation,
we study the ability of CNNs to recognise and respond to such local features in the single column simulations.

Table 1 illustrates the structures and parameters of our neural networks. Specifically, we have designed five neural networks, including two feedforward neural networks and three convolutional neural networks (CNN). Model A and Model B are implementations of feedforward neural networks with different numbers of layers and of neurons in each layer. Model C is a simplified CNN implementation based on previous work (Simonyan and Zisserman, 2014). The stride of convolutional filters
is set to 1 so that the convolutional filters go through the input array with 1 element each step. We have not applied any padding

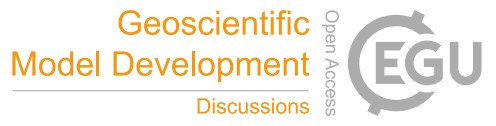

| Model Name | Model Structure | | | | | | Number of Parameters |
|---|---|---|---|---|---|---|---|
| Model A | input-60x4 | fc-512 | fc-1024 | fc-512 | output-60 | | 1079808 |
| Model B | input-60x4 | fc-512 | fc-1024 | fc-2048 | fc-1024 | fc-512 | output-60 | 5274112 |
| Model C | input-60x4 | conv-3x3-128 | conv-3x3-256 | fc-512 | output-60 | | 457856 |
| Model D | input-60x4 | conv-3x3-128 | conv-3x3-256 | conv-3x3-256 | fc-512 | output-60 | 1047680 |
| Model E | input-62x6 | conv-3x3-128 | conv-3x3-256 | fc-512 | output-60 | | 457856 |

**Table 1.** Neural network models used for predicting RTs. "fc-X" represents a fully connected layer with X number of neurons. "conv-YxY-X" represents a convolutional layer with X number of YxY filters.

to the input. The convolutional filters are the classic 3x3 filters. Model D is a variant of Model C with one more convolutional layer. Model E has the same neural network structure as Model C. The only difference is that model E has padded the input with an edge of zeros to emphasize on edges of the input.

In addition to the above models, we also evaluate a variant of model E denoted as model F. Model F is based on Model E but with a fixed pressure grid. This means that Model F does not take pressure values as input, and interpolates air temperature and humidity from model levels onto a fixed, time-invariant pressure grid. While this configuration reduces the dimensionality of the input, it requires extrapolation of the ERA-Interim data or the calculated/predicted RT to the fixed grid. Specifically, the inputs of a sample profile are B-spline extrapolated according to a fixed pressure grid. We extrapolate the air temperature and humidity values onto the fixed pressure grid based on the profile's pressure range. The inputs corresponding to the rest of the pressure levels are set to 0. After running through model F, the RTs on the static pressure grid are B-spline interpolated back to the original pressure levels, which are the final results. It is important to mention that we constructed the static grid using 15 equally spaced pressure levels from 1 to 500 Pa, another 15 equally spaced pressure levels from 550 to 50000 Pa, and 30 equally spaced pressure levels from 50300 to 103000 Pa. We made this design choice by observing the distribution of the ERA-Interim data to ensure that our fixed grid encompasses most common pressure profiles in order to achieve a better accuracy on the extrapolation and interpolation. We used model F to run the single-column model simulation presented below, which employs a fixed pressure grid.

## 3.2 Model training

We trained our five NNs with two datasets, resulting 10 different models. The first dataset is the aforementioned ERA-Interim dataset, namely, $Dataset_1$. The second dataset is the augmented dataset, i.e., $Dataset_2$, in order to generalize the model to a wider operational region beyond $Dataset_1$. Each neural network is trained using the training dataset of either $Dataset_1$ or $Dataset_2$ and validated against either $Dataset_{1.val}$ or $Dataset_{2.val}$.

Each model was trained with 30 epochs under a batch size of 128 starting with a learning rate of 0.001, which then exponentially decays every 10 epochs with base 0.96. This setup was empirically obtained while we observe that all models have converged after the training. Mean squared error is used as the loss function in all models. Parametric Rectified Linear





185 Units (pReLUs) (He et al., 2015) are used as activation functions in all models since PReLUs is able to resolve the problem of vanishing gradient during model training. Adam optimizer (Kingma and Ba, 2014) is employed to compute the gradients.

We present the evaluation results regarding the performance of these models in the next section.

## 4    Evaluation

### 4.1    Evaluation Setup

190 We prepared two datasets, i.e., $Dataset_{1.val}$ and $Dataset_{2.val}$, to evaluate our neural network models. $Dataset_{1.val}$ is used to evaluate the accuracy of the trained models with realistic future data. $Dataset_{2.val}$ is used to evaluate the generality of the trained models as it contains profiles that are perturbed versions of the ERA Interim data.

### 4.2    Prediction Accuracy

We use vertical level-wise root-mean-squared errors (RMSE) to compare our NN generated radiative cooling rates with those 195 generated by the RRTMG algorithm. Figures 2-5 present results for the different NN models. The RMSE is calculated by taking the difference between NN- and RRTMG-calculated radiative cooling profiles.

Figure 2 presents the RMSE when the NN models are trained using $Dataset_1$ and validated against $Dataset_{1.val}$. These experiments are performed to evaluate the capability of different NN models to predict RT when the atmospheric profiles are sampled from the ERA-Interim dataset itself. We see that a simple 3 layer feedforward neural network (Model A) is able to 200 predict heating rates with a median RMSE of lesser than 0.01 K/day across all pressure ranges. The performance does not improve when more layers of directly connected neurons are added, as shown by Model B. We observe significant RMSE improvement while using CNNs (Model C,D,E). However, the performance differences among these three CNN models are not substantial except the RMSEs near the surface, which tend to have higher variability as shown in the statistics in Figure 1. Since surface radiation is particularly important to the climate, efforts have been made to minimize its prediction error. The 205 input matrix of Model C are padded with zeros in order to allow convolutional filters to put equal emphasis on the edges values as the middle ones (Innamorati et al., 2018). This creates Model E, which shows much better prediction accuracy on the bottom and top pressure levels.

Figure 3 presents the RMSE when the NN models are trained using $Dataset_2$ and validated against $Dataset_{1.val}$. In this experiment, we examine whether it is possible to expand the operational region of the NN models without compromising on 210 their performance on the ERA-Interim dataset. Comparing to Figure 2, we see that the increased generality comes at the cost of roughly doubled RMSE across all models.

The improvement on generality is suggested by the results shown in Figure 4 and Figure 5 when the NN models are trained using $Dataset_1$ or $Dataset_2$ and validated against $Dataset_{2.val}$. The RMSE increases by almost 100 times across all models trained with $Dataset_1$ and validated against $Dataset_{2.val}$ (Figure 4) when compared to their validation against $Dataset_{1.val}$ 215 (Figure 2). This suggests that that models trained with $Dataset_1$ cannot really generalize to predict heating profiles from





$Dataset_{2.val}$. On the other hand, the RMSE increases 10 times when the models are trained using a wider range of data, i.e., $Dataset_2$, as shown in Figure 5. This is mainly because that the model needs to cover a larger operational region.

When trained on $Dataset_1$ and validated against $Dataset_{2.val}$ (Figure 4), the RMSE in Model B is significantly higher. We believe that this is because of over-fitting. With more parameters in Model B and the nature of feedforward NN (Goodfellow
et al., 2016), it is more likely that parameters are deeply coupled with patterns observed in $Dataset_1$ and have larger errors while evaluating against $Dataset_{2.val}$.

Model F displays significant errors on both edges of the pressure levels. This is due to extrapolation errors. Specifically, if the lowest pressure level in an atmospheric profile is lower than the lowest pressure level of the fixed grid, the profiles need to be extrapolated. The same issue arises with the highest pressure levels as well. Thus, the errors in model F are mainly due to
these extrapolation based artifacts rather than an issue with the training itself. In fact, this model provides the most stable time integration of the single column model.

In the above evaluations, we have shown that CNN-based models achieve much lower prediction RMSE than feedforward NN models. However, in the next section, we show that CNN-based models tend to have much slower prediction speed, i.e., less speedup as compared with the feedforward models.

**4.3 Prediction speed**

In this section compare the computation time of RRTMG and NN models using GPUs and CPUs. These performance evaluations have been performed using Intel Xeon CPU E3-1230 v5 @ 3.40GHz, Nvidia GTX 1060 GPU with 6GB of onboard memory and Nvidia GTX 1080 GPU with 8GB of onboard memory. Both GPUs we use are commodity hardware, and are easily available in the market.
Table 2 summarizes the speedups using NN models to predict RT as compared to RRTMG. The calculation time of NN code and RRTMG code are profiled using the python line_profiler based on cProfile. Since RT calculations are embarrassingly parallel, we are able to use batch predictions in our NN models. The overall results show that the larger the batch size, the larger the speedup observed as long as the CPU or GPU memory is sufficient. In other words, the calculation of $M$ radiative heating profiles is faster than $M$ times the time taken to predict one such profile. This is because of the efficiency of matrix
multiplications in NNs while conducting NN forward pass in batches. We note that such a speedup is not possible in a physics-based RT scheme since the calculation of RT for an arbitrary atmospheric profile cannot be expressed as a simple matrix multiplication.

The results show that using only the Xeon CPU, NN model A and B are able to achieve speedups up to 10.88x and 2.82x respectively using a batch size of 1024. When using GTX 1060, we are able to achieve speedups of 123x in model A, 61x
in model B and 2.8x to 4.5x in CNN-based models (C,D,E). With GTX 1080, which has a larger memory and a faster clock speed, we observe speedups up to 370x in model A, 123x in model B and 4.4x to 7.7x in CNN-based models (C,D,E).

The results indicate that if the prediction accuracy of Model A is sufficient for a climate simulation, it will provide the greatest calculation speedup either using CPU or GPU. Since NNs with comparable or worse accuracy have been used for simulations ranging from months to years (Krasnopolsky et al., 2008; Pal et al., 2019), Model A is a promising candidate for modelling





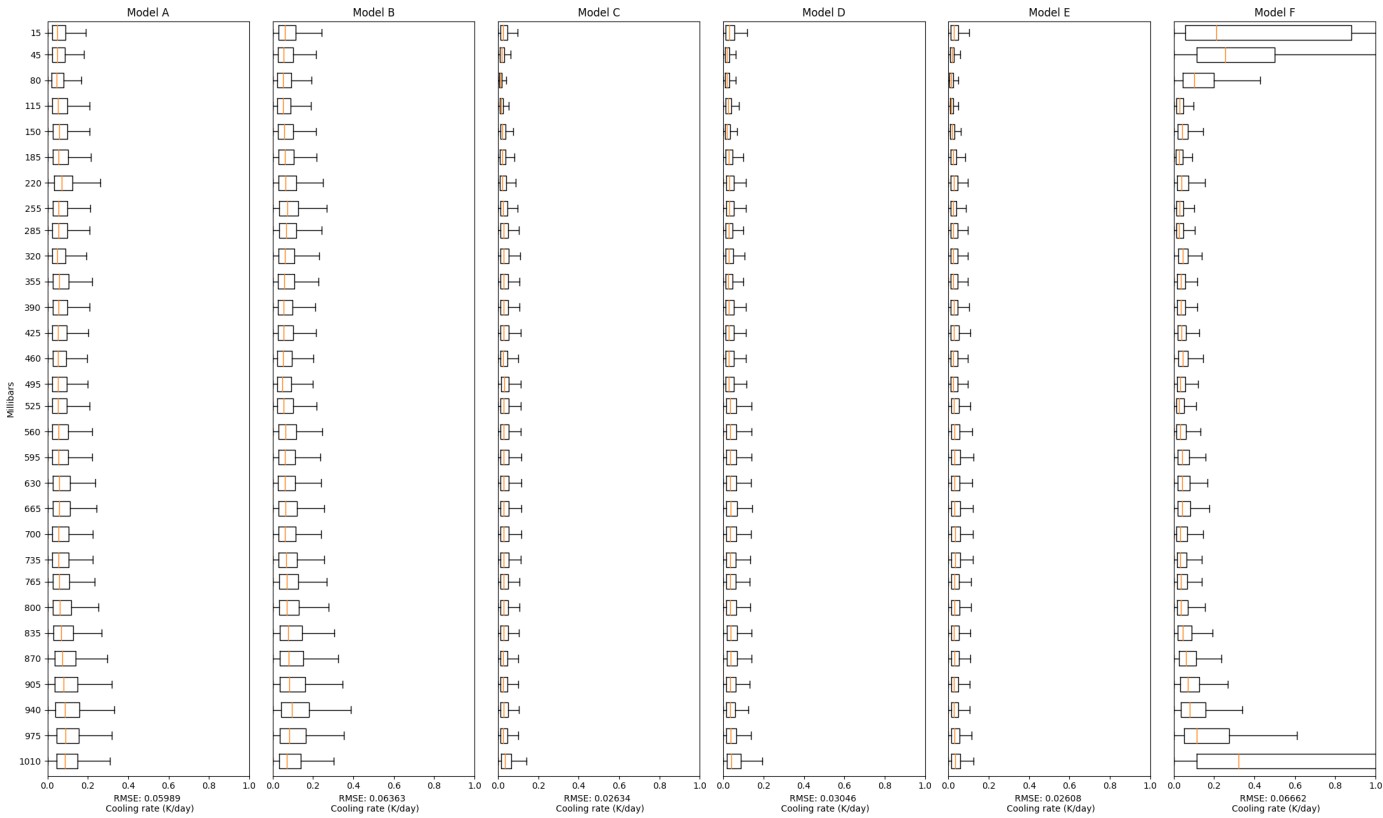

**Figure 2.** Models are trained with $Dataset_1$ and evaluated against $Dataset_{1.val}$. The models have 30 linearly spaced levels between 0 and 101300 pascals. The boxes in the plots present the boxplot of the RT MSEs at every level. Specifically, the boxes describe the 25 (Q1), 50 (Q2) and 75 (Q3) percentile of the MSEs while the two whiskers extend from the edges of box to 1.5 times the interquartile range (Q3-Q1).

applications since similar performance gains using a full RT code seems to require a complete rewrite for GPUs (Price et al., 2014; Mielikainen et al., 2016). For simulations requiring a higher accuracy, Model C provides significant speedups even if a normal GPU is available on the platform.

### 4.4 Single column model simulation

To explore the ability of the NN model to generalise to new situations, we compare the climate of a single column model when
RRTMG is replaced by the NN model F (see previous section for a description of model F). The single column model uses a diffusive boundary layer (Reed and Jablonowski, 2012), a slab surface of 50 meters thickness which behaves like an oceanic mixed layer, the RRTMG shortwave component, and the Emanuel convection scheme (Emanuel and Zivkovic-Rothman, 1999). The model has no seasonal or diurnal cycle. Carbon dioxide concentration is fixed at 300 ppm, and a fixed ozone concentration is prescribed using an observed tropical profile. The model uses pressure as the vertical coordinate and has 60 equally spaced





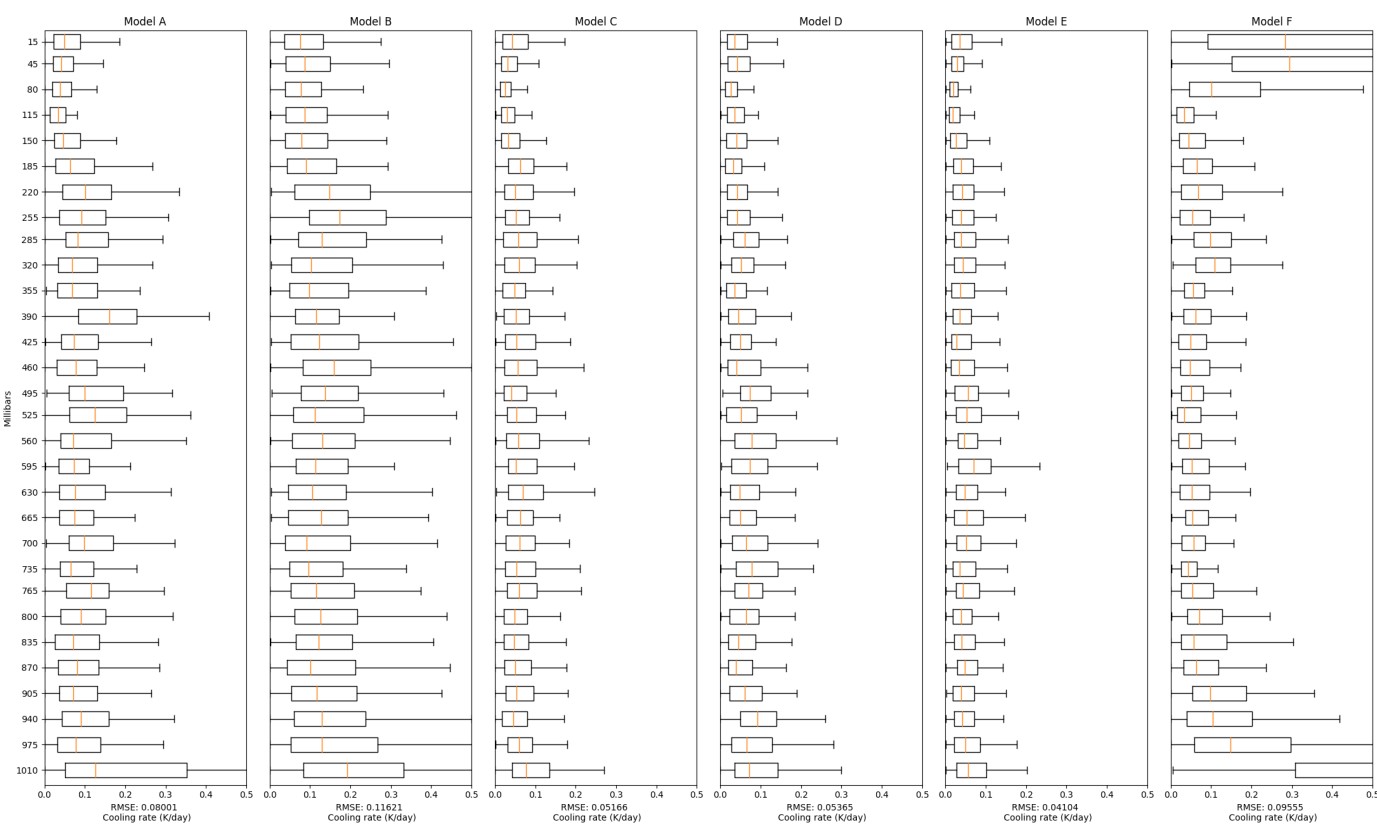

**Figure 3.** Models are trained with $Dataset_2$ and evaluated against $Dataset_{1.val}$

| Baseline RRTMG | 0.37 ms | | | | | | | | | |
|:---:|:---:|:---:|:---:|:---:|:---:|:---:|:---:|:---:|:---:|:---:|
| NN Model Name / Hardware | Xeon CPU E3-1230 | | | GTX 1060 | | | GTX 1080 | | | |
| NN Batch size | 64 | 256 | 1024 | 64 | 256 | 1024 | 64 | 256 | 1024 | 4096 |
| Model A | 5.87 | 10.28 | 10.88 | 18.50 | 61.67 | 123.33 | 16.08 | 61.67 | 123.33 | 370.00 |
| Model B | 1.87 | 2.74 | 2.82 | 13.70 | 37.00 | 61.67 | 14.80 | 46.25 | 74.00 | 123.33 |
| Model C | 0.14 | 0.14 | 0.14 | 3.19 | 4.11 | 4.57 | 4.25 | 5.52 | 7.40 | 7.71 |
| Model D | 0.11 | 0.11 | 0.11 | 2.52 | 3.03 | 3.33 | 3.52 | 4.25 | 5.44 | 5.52 |
| Model E | 0.09 | 0.09 | 0.09 | 2.16 | 2.59 | 2.82 | 2.98 | 3.67 | 4.63 | 4.40 |

**Table 2.** Speedups when using NN models to predict RTs comparing to calculating RTs using RRTMG. Result for RRTMG is shown for the calculation of a sample in units of milliseconds. Results for NN models are shown as speedups in different batch sizes comparing to the RRTMG calculation speed on the Xeon CPU.

vertical levels between 1013.2 hPa and the model top value at 0 hPa. The model time step is 10 minutes. The tendencies from the various components are stepped forward in time using a third order explicit Adams-Bashforth scheme.



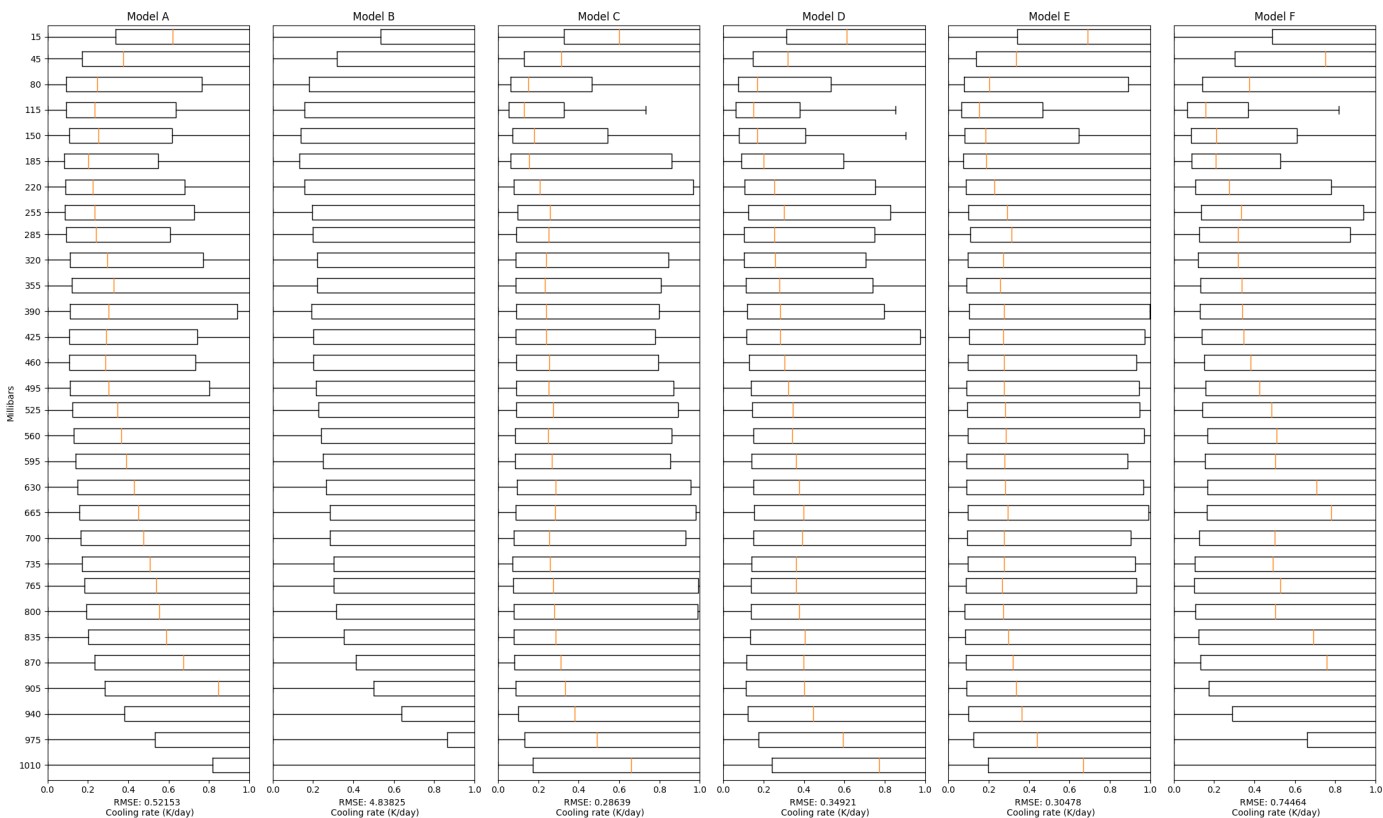

**Figure 4.** Models are trained with $Dataset_1$ and evaluated against $Dataset_{2.val}$

The model is initialised with a dry, isothermal state. We use RRTMG's longwave component to drive the model until the RMS error between the RRTMG calculated longwave heating rates and those predicted by model F falls below a threshold of 0.5 K/day. Once the errors falls below this value, model F takes over and RRTMG's longwave component is never used again for the rest of the simulation (shortwave radiation is computed using RRTMG throughout). The switch from RRTMG to model F happens after around 14 days of simulation. This simulation is denoted as "RadNet" in Fig. 6. Another simulation continues to use RRTMG longwave radiation until the end of the simulation and is denoted as "RRTMG" in Fig. 6. As discussed subsequently, the RadNet simulation has a bias in the stratosphere and the temperature profile of the top three levels is constrained to the RRTMG simulation to prevent the simulation from blowing up. Both simulations are run for 2100 days and equilibrium is reached around 1600 days, with constant temperature and humidity profiles afterwards.

Within the troposphere, both simulations show a realistic moist-adiabatic temperature profile and are in reasonable quantitative agreement. However, there are substantial differences in the stratosphere, and the equilibrium position of the tropopause seen in Fig. 6c in the RadNet simulation is higher by around 50 hPa as compared to the RRTMG simulation. This is because Model F has a cooling bias in the upper atmosphere as seen in Fig. 6d, which makes it convectively unstable and therefore the





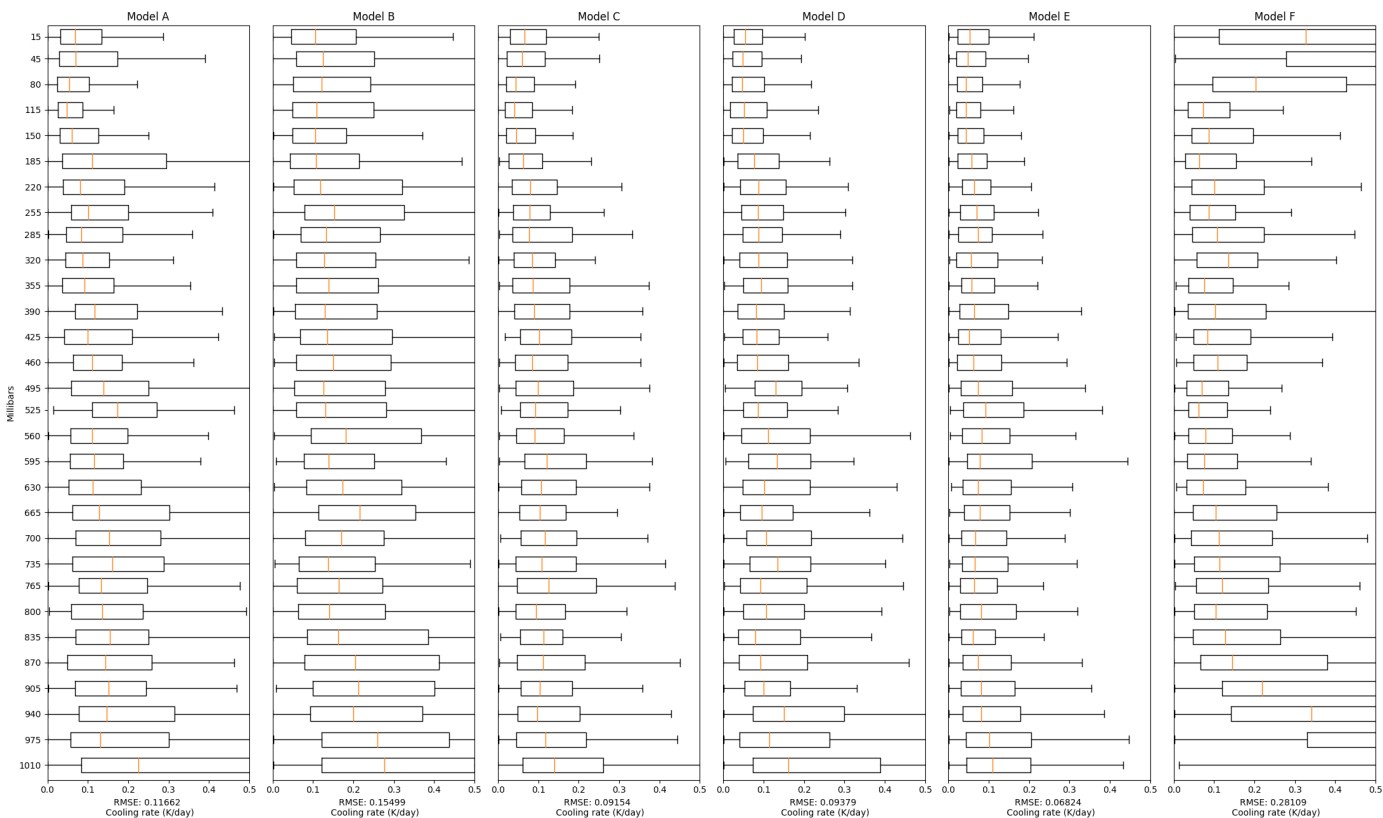

**Figure 5.** Models are trained with $Dataset_2$ and evaluated against $Dataset_{2.val}$

tropopause shifts upward. The tropospheric temperature profiles are identical since they are set by the convective parameterization in such convectively unstable situations.

As the boundary layer fluxes water vapour into the column from the surface, the atmosphere becomes opaque to longwave radiation in the lower levels and therefore the longwave cooling is strongest in the level just above the moist, opaque part of the atmosphere. Figure 6d shows that the cooling peak predicted by model F has a smaller magnitude and is located lower in the atmosphere. The lower cooling rate peak predicted by the NN results in the slower evolution of the RadNet simulation as compared to the RRTMG simulation, resulting in the difference in height between the two simulations (the cooling peak rises over time as the convection tries to eliminate the instability produced by radiative cooling). The cooling peak in the RadNet simulation is situated close to the location of the strongest gradient in water vapour (where the atmosphere transitions from being opaque to transparent to longwave radiation), which is physically accurate. The differences in magnitude are larger slightly earlier in the simulation, where the atmospheric profiles are quite unlike the profiles in the training dataset. It seems unlikely that neural nets can predict such "spiky" profiles correctly since the predicted results tend to be smooth in general.



**Figure 6.** Comparison of the vertical profiles of temperature (first row), longwave heating rates (second row) and specific humidity (third row) for the RadNet and RRTMG simulations at three different times.





However, the Radnet predicted profiles provide sufficient cooling to make the atmosphere convectively unstable and eventually mix the entire troposphere of the model.

The NN has a systematic warm bias in the lowest layer of the model, which may be linked to the interpolation errors
discussed previously for model F. This warm bias results in a slightly warmer surface temperature ($\sim 0.5K$) in the RadNet simulation as seen in Fig. 6c. The warmer profile supports a larger amount of water vapour, and the RadNet simulation has a moist bias in the lower troposphere as well.

We see that small systematic errors in the predicted heating rates can have a non-trivial effect on the simulated climate in a single column model, especially in the upper layers of the atmosphere. In particular, errors in radiative heating near the
tropopause can dramatically change the structure of this part of the atmosphere. The neural network tends to cool the upper atmosphere a little more, making it more convectively unstable and pushing the convection and tropopause higher.

To verify the accuracy of the predicted heating profiles, we use the atmospheric profiles from the RadNet simulation to drive the RRTMG longwave component. The NN and RRTMG heating profiles generated are presented in Fig. 7. The heating profiles predicted by the NN are fairly accurate, especially in the later parts of the simulation when the atmospheric profiles are similar
to those in the training sample space. The NN predicts the location of the cooling peak accurately even when the atmospheric profiles are unlike those in the training sample space, though it underestimates the magnitude. RRTMG produces fairly noisy heating profiles in the stratosphere, reflecting the noisy temperature profile simulated by the NN. The noisy stratospheric temperature profile appears to be a result of the fact that the training data for model F was generated using atmospheric profiles that had additional noise added to them, which results in noisy heating profiles used for training.

## 5   Discussion and Conclusions

Radiative transfer was probably among the first climate model components that neural network models aimed to replace in climate simulations. The evolution of NN models has paralleled the evolution of NN architectures themselves, with initial attempts using shallow networks while recent attempts (including our own) using deep networks. Since both shallow and deep networks seem to perform reasonably well in model simulations (Krasnopolsky et al., 2008; Pal et al., 2019), the question of
which type of architecture is more suitable inevitably arises.

Recent work in NN theory suggests that the mathematical structure of deep neural networks (a series of linear and non-linear operators applied sequentially) is especially suited to capture functions which can be expressed as the composition of other functions (Mhaskar and Poggio, 2016; Lin et al., 2017). Radiative transfer conforms to this structure very well: the total radiative heating rate is the sum of heating rates in each spectral band, and the heating rate in each spectral band requires
the calculation of absorption coefficients at each model level, each independent of the other. The two-stream approximation and the independent column assumptions introduce additional locality and symmetry requirements, constraining the problem further. This mathematical structure suggests that deep neural networks are a natural choice to approximate RT. Furthermore, the presence of highly localised scattering and absorbing substances such as clouds and water vapour suggest that RT might benefit from a NN structure which is sensitive to localised patterns. This suggests that convolutional NNs might be a better





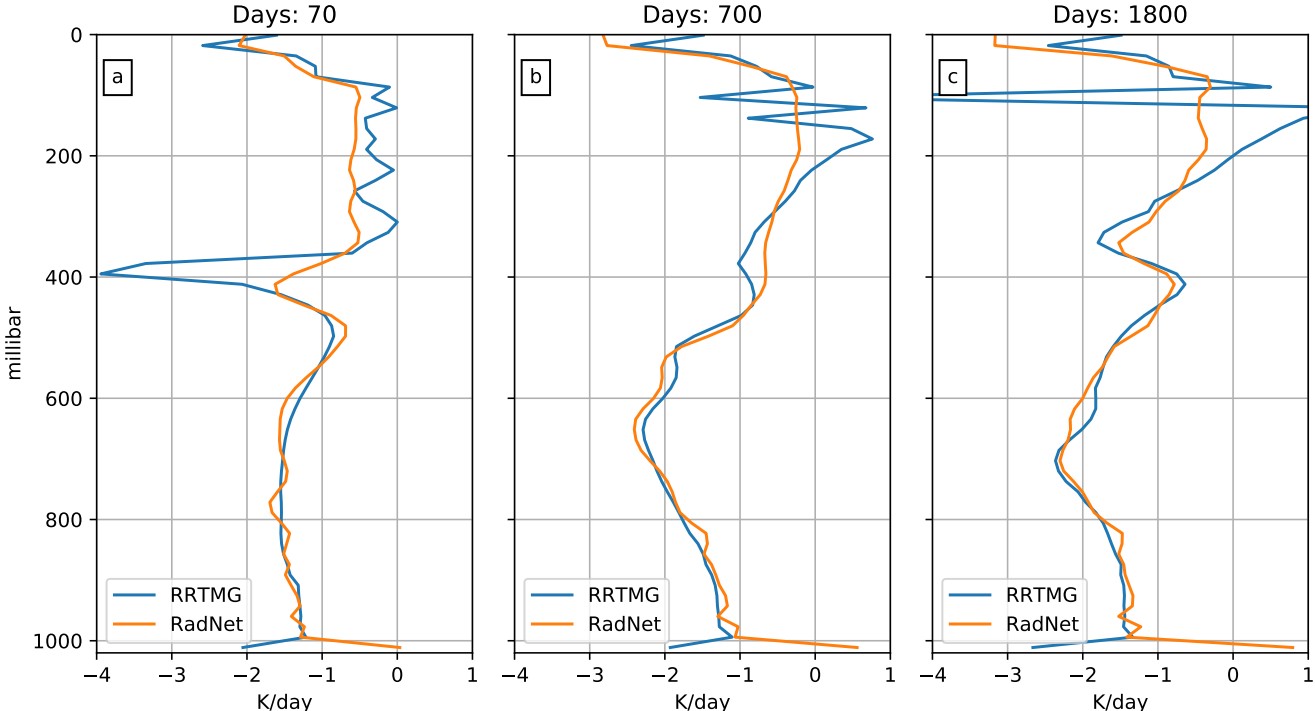

**Figure 7.** Comparison of the vertical profiles of longwave heating rates predicted by the NN and RRTMG for atmospheric profiles from the RadNet simulation.

model for RT, and our results confirm this. However, our results also show that using convolutional NNs reduces performance by 50-100 times as compared to feedforward NNs with only a marginal increase in accuracy. Thus, within our evaluation setup, deep feedforward NNs present the best compromise between accuracy and performance.

The ability of NNs to generalise to unfamiliar atmospheric profiles seems to be limited as suggested by the cases where the NNs were validated on the perturbed dataset and the single column model comparisons. These results bring to question the applicability of NN based radiative transfer in research configurations where perturbations to the model state or evolution to a wholly new climate state is routinely performed. Thus, NNs seem to work best in an "operational mode" where the state of the climate or weather prediction model is not expected to change dramatically as compared to the training set. The approach of adding of noise to improve NNs' ability to generalise beyond the training sample has a long history (Sietsma and Dow, 1991). However, our results show that adding noise to the training dataset results in noisy temperature profiles in simulations, especially in the stratosphere where the temperature profile is closer to pure radiative equilibrium.

The dramatic performance gains when using commodity GPUs makes the use of NNs all the more attractive given that most future high-performance computing configurations will include both GPUs and CPUs. NNs allow batching of multiple atmospheric profiles during matrix multiplications, which allows large performance gains. Such batching is not feasible for an actual RT calculation, and each atmospheric profile has to be handled individually. This may be the reason why a complete



rewrite of RRTMG for GPUs gives very similar performance gains as what we have achieved in our setup using NNs (Price et al., 2014; Mielikainen et al., 2016).

Another method to assess the ability of NNs to generalise is to actually build a climate model which includes the NN as a component. Since single column models have no diffusion built-in and cannot transport energy horizontally, we believe that they constitute a tougher test case for NNs as compared to GCMs. The lack of dynamics also makes the results easier

to interpret. In our test case, we see that the errors in prediction by the NN has a larger impact in the stratosphere than the troposphere due to the tight control of the tropospheric lapse rate by moist convection. The initial atmospheric profile – dry and isothermal – is quite different from the profiles in the training sample space. While the errors in the initial part of the simulation are larger, the NN predicts physically realistic heating profiles with slight differences in location and magnitude. Such physically plausible behaviour in situations quite different from those the NN was trained on gives us confidence that NNs

can indeed be used as climate model components in the future. However, it is clear that better strategies for data preparation, selection of NN architecture and testing trained NNs are required to improve NN performance and enable scientists to interpret their impact on climate model simulations.

*Code availability.* The code used for training RadNet and the Jupyter notebook which simulates the single column model are available at http://doi.org/10.5281/zenodo.3609222

*Author contributions.* YL designed, trained and validated the neural networks, prepared the input data and participated in design of the project and writing the paper. RC participated in design of the project and writing the paper. JMM wrote the climt code to generate the data, interface RadNet to the single column model, analysed the simulations and participated in writing the paper.

*Competing interests.* The authors declare no competing interests.

*Acknowledgements.* All authors were funded by the Swedish e-Science Research Centre (SeRC).



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
