# Peer review of "RadNet 1.0: Exploring deep learning architectures for longwave radiative transfer"

_Geoscientific Model Development, 2019_

## Referee Comment (RC1) · Anonymous Referee #1 · 19 Feb 2020

Summary

In this paper the authors use a neural network to emulate a radiation scheme used in GCMs. They use reanalysis states to train networks with several architectures and find that the radiative heating rates were accurately reproduced. They then couple the emulator to a single-column simulation and find that the climate of the neural net model differs in some ways from the reference climate.

This paper represents a valuable contribution to the growing literature of machine learning parameterizations and should be of interest to a range of potential readers. The paper is well written and the analysis is thorough. I do have a few comments about the methodology and interpretation of the results though which are listed below.

[Figure]

General comments

1. Study setup You have chosen to use ERA profiles as input which of course gives you realistic profiles. However, this comes with two major limitations: a) You have to manually create out-of-sample profiles (Dataset 2) which are potentially unrealistic (see comment below). b) You are limited to single-column model experiments for coupled verification. While I don't expect you to do this for this revision, I feel like you have the chance for a more insightful experimental setup using CliMT. You could have setup a global CliMT simulation that is somewhat realistic and used the RRTMG output from that simulation as training data. Then you could have also ran the same experiment with increased SSTs for a more realistic climate change experiment. Lastly, you could have used your RadNet in directly in the global simulation for a 3D coupled experiment. I think the results from such an experiment could be really insightful for the ML parameterization community.

2. Training data estimation for line-by-line computation As you mentioned in the paper, ideally one would use line-by-line radiation computations to train from. Since line-by-line is really expensive you would be limited in the amount of training data you can generate. Would it be possible to make an estimate of what a realistic amount of line-by-line data would be and whether this amount of data would be enough to train your networks

3. Neural network architectures One of the key goals of this paper is to compare different NN architectures, particularly fully connected versus convolutional. You present 5+ different architectures. I have several questions about these:

3.1 Model E is the same as Model C but with zero-padding but why do the number of channels also increase in the table from 4 to 6? I assume this is an error since the number of parameters also stays the same.

3.2 You could have tried a fully convolutional network by using zero-padding on each layer. That way you don't need the final fully connected layer, which means a lot fewer parameters. I would be interested to see how such a network would perform.

3.3 However, one problem I see with CNNs in this context is that they are based on translational invariance i.e. the computations are the same along the entirety of the vector. But with non-uniform grids like the one you are using is there any reason to assume that translation invariance should hold?

3.4 Another architecture you could try is a Resnet using skip connections.

3.5 You mention you use PReLUs to avoid vanishing gradients. Did you observe vanishing gradients for normal ReLUs? That would surprise me since your networks are not very deep.

3.6 Did you observe overfitting (train MSE < valid MSE)? Your validation dataset is from 2015-2016 while your train dataset is from 1979-1985, which means that the base state is probably changing. This will make it hard to check for overfitting since even with a model that does not overfit one would expect the train loss to be lower. A better test of overfitting would be to take a random subset of the train dataset. The reason I am interested in this is that I am surprised a bigger network (B vs A) did not give you a better score. If you are not yet overfitting, bigger networks should always result in better losses, or am I missing something? You mention overfitting when talking about the generalization experiments but that still doesn't explain why the skill for Dataset 1 wouldn't be better for model B.

——-

Specific comments

Line 56: You could also mention this recent paper: https://arxiv.org/abs/2001.03151. Also this follow up paper highlights some interesting issues: https://onlinelibrary.wiley.com/doi/abs/10.1029/2019MS001711

Line 90: Typo "colummn"

Line 102: I think it would be useful at this point to mention the horizontal and vertical resolution of ERA.

Line 105: So the size of your training dataset is around 12 million? Please mention this explicitly?

Line 110: If I understand correctly there is no vertical correlation in your perturbation, right? Are the resulting profiles therefore "unrealistic"? How do you think this affects the network's generalization? Do you think that your results are representative of the sorts of perturbations you would experience in real climate change (increased T)?

Line 150: Actually, convolutions are a form of regularization by introducing an architectural constraint. In fact, a convolution layer is simply a subset of a fully connected layer with shared weights (https://medium.com/impactai/cnns-from-different-viewpoints-fab7f52d159c). But yeah, in effect you end up focusing on local features.

Table 1: I think it would be helpful to list all the Models (including F) in the table.

Data: Great that you included an example file in the repo. But maybe you could also add instructions on how to download the full dataset you used.

Code: Kudos for including code. I looked at the implementation of the networks. Maybe just a recommendation: I think the code could have been a lot easier and clearer if you used Keras instead of plain TF. Especially with TF 2 its now very easy to take all the precoded parts of Keras and implement your own layers, etc. using TF. All the manually coded layers in your code (matmul, etc.) make it very hard to follow when a simple Dense() would suffice. This would make the code easier to read for others.

---

## Referee Comment (RC2) · Anonymous Referee #2 · 20 Feb 2020

Interactive comment on"RadNet 1.0: Exploring deep learning architectures for long-wave radiative transfer" byYing Liu et al.

Anonymous Referee

The paper explores the use of neural networks architectures to model radioactive transfer in global circulation models. The paper is interesting and well written. Authors carried out thorough experiments and the conclusions about the use of NN in the field are interesting. Additionally, the source code for the implementation is publicly available, which is really useful. I have a few concerns about the experiments and how the performance is calculate.

- Speedups of table 2 are obtained by comparing the computational time of RRTMG and NN. I suppose that the baseline time for RRTMG is obtained in the Xeon CPU, and all the values are computed using this baseline. The objective of the paper is to evaluate the advantage of replace a RRTMG for a NN. To do that, a comparison in term of execution time is done. But, to be fair, the comparison of the NN should have been done with a RRTMG implementation for GPU. In the conclusions, authors say that "a complete rewrite of RRTMG for GPUs gives very similar performance gains as what we have achieved in our setup using NNs (Price et al., 2014; Mielikainen et al., 2016)". You should include evidences to support this affirmation. For example, in Mielikainen et al. author state that their implementation provides a speedup of 202x on a Tesla K40 GPU, compared a single-threaded Fortran counterpart running on Intel Xeon E5-2603 CPU. A higher speedup (370x) is indicated in Table 2, for NN model A and a batch size of 4096 running in GTX 1080. Both GPUs architectures are very different, so result are not directly comparable. In fact, single precision floating point (float32) performance is higher in the GTX 1080 that in the Tesla K40 (it's the opposite for double precision).

- Related with the previous question, Intel Xeon CPU E3-1230 v5 is a quad-core processor (8 virtual cores). Is the execution of RRTMG single or multi-threaded? This should be indicated.

- More details should be included about how the time is measured. cProfile is used for calculate the execution time, but with times of only 0.31 ms, as indicated in table 2, the accuracy of the measure is not guaranteed, as a typical time granularity on Unix is 1 ms. Alternatives, as the time Python module, could provide more accurate measurements. Additionally, information about how many measurements have been done and how the final values have been computed should be included.

Additional comments

- Introduction: To include a specific section with the related work, instead of including it in the introduction, could be beneficial

[Figure]

- A recent work describing other GPU-based accelerating methods for the RRTMG_LW could be analyzed (https://www.mdpi.com/2076-3417/10/2/649)

- Section 2.2. Is it realistic the perturbed dataset as has been created? Can be these values representative of a real situation?

- Section 4.4. In the single column model simulation, why the NN model F is used? Why not use model A or model C?

- The developed code used TensorFlow, this should be indicated in the paper along with the version employed. It could be also interesting to describe the benefits of using this library in comparison with others, eg PyTorch.

───────────────────────

---

## Author Comment (AC1) · 14 Apr 2020

article [utf8]inputenc enumitem color url

**Reply to the Reviewer 1 comments for: RadNet: Exploring deep learning architectures for longwave radiative transfer**

We thank the reviewer for their comments.

**1   General Comments**

1. ```
Study setup You have chosen to use ERA profiles as input
which of course gives you realistic profiles.  However,
this comes with two major limitations:  a) You have to
manually create out-of-sample profiles (Dataset 2) which
are potentially unrealistic (see comment below).  b) You
are limited to single-column model experiments for coupled
verification.  While I don't expect you to do this for
this revision, I feel like you have the chance for a more
insightful experimental setup using CliMT. You could have
setup a global CliMT simulation that is somewhat realistic
and used the RRTMG output from that simulation as training
data.  Then you could have also ran the same experiment
```

with increased SSTs for a more realistic climate change
experiment.  Lastly, you could have used your RadNet
in directly in the global simulation for a 3D coupled
experiment.  I think the results from such an experiment
could be really insightful for the ML parameterization
community.

**Response:** We thank the reviewer for these suggestions. We would like to clarify that:

a) Our validation dataset was not specifically created with a climate change scenario in mind.  Rather, our aim was to explore the response of NNs when perturbations are added to the input profiles.  Radiative heating profiles can be fairly noisy since optical parameters can change quite drastically in the vertical, and these changes need not be highly correlated.  For example, the presence/absence of clouds, hydrometeors, aerosols, horizontal advection of water vapour can cause localised perturbations which are not strongly correlated vertically. That being said, we agree that our validation dataset represents an extreme example of this behaviour. We have added a paragraph in Section 2.2 to explain the reasoning that went into creating the perturbed dataset.

b) There is nothing in our setup which limits our simulations to a single column. This is because radiative transfer is assumed to have no contribution from adjacent columns (as we have described in Lines 91–92.).  Our motivation to use a single column model for validation has been described in the manuscript in line 350: it allows us to study the evolution and possible errors in greater detail.

We thank the reviewer for their suggestions regarding the proposed experimental setup, which are natural extension to the work presented here, and we look forward to following this up in future work.

**Lines Changed:** 129–133

2. `Training data estimation for line-by-line computation`
`As you mentioned in the paper, ideally one would use`
`line-by-line radiation computations to train from. Since`
`line-by-line is really expensive you would be limited in`
`the amount of training data you can generate. Would it be`
`possible to make an estimate of what a realistic amount of`
`line-by-line data would be and whether this amount of data`
`would be enough to train your networks`

   **Response:** The dataset used to train the network in this paper is around 12 million training samples and 3.5 million validation samples. Thus, the total dataset size generated is around 15 million samples. RRTMG requires 0.37 ms to generate a sample as per our profiling results (Table 1). The total time to generate all samples amounts to around 1.5 hours. The performance of a line-by-line radiative transfer code is dependent on the spectral resolution it uses; However, it is typically around 2-3 orders of magnitude slower than correlated-k codes (see Table 2 in https://doi.org/10.3390/rs11090994). Using a 10-core machine (for example) to generate the samples, we would require it to run continuously for around 2 weeks to generate the required samples. This estimate does not take into consideration any memory-related constraints that may arise.

3. `Neural network architectures One of the key goals of`
`this paper is to compare different NN architectures,`
`particularly fully connected versus convolutional. You`
`present 5+ different architectures. I have several`
`questions about these:`

   (a) `Model E is the same as Model C but with zero-padding`
   `but why do the number of channels also increase in the`
   `table from 4 to 6? I assume this is an error since the`
   `number of parameters also stays the same.`

**Response:** The channel number is correct. The increase is caused by the zero-padding. However, it is good that you have pointed out the parameter numbers. We made an error in calculating the numbers. Here are the correct number of parameters with calculation steps. The number of parameters are also supported by the speedup presented in the later section.

Model A: $240 * 512 + 512 * 1024 + 1024 * 512 + 512 * 60 = 1202176$

Model B: $240 * 512 + 512 * 1024 + 1024 * 2048 + 2048 * 1024 + 1024 * 512 + 512 * 60 = 5396480$

Model C: $3 * 3 * 128 + 3 * 3 * 128 * 256 + 56 * 1 * 256 * 512 + 512 * 60 = 7666816$

Model D: $3 * 3 * 128 + 3 * 3 * 128 * 256 + 3 * 3 * 256 * 256 + 54 * 1 * 256 * 512 + 512 * 60 = 7994496$

Model E: $3 * 3 * 128 + 3 * 3 * 128 * 256 + 58 * 2 * 256 * 512 + 512 * 60 = 15531136$ We have also made the appropriate changes in the manuscript.

**Lines Changed:** Table 1, Row 5

(b) ```
You could have tried a fully convolutional network by
using zero-padding on each layer.  That way you don't
need the final fully connected layer, which means a
lot fewer parameters.  I would be interested to see how
such a network would perform.
```
**Response:** We do not think that adding padding to all conv layers will act the same as fully connected layers. The way to construct a fully connected layer using conv layer is to use conv filters the same size as the input layer with stride 1. So that each conv filter computes the input only once covering all vectors then the number of conv filters aggregates, e.g. 128. Then, it will act like 128 neuron fully connected layer.

(c) ```
However, one problem I see with CNNs in this context
is that they are based on translational invariance
i.e.  the computations are the same along the entirety
of the vector.  But with non-uniform grids like the
one you are using is there any reason to assume that
translation invariance should hold?
```

**Response:** We are unsure what the reviewer means here. We presume they are referring to the fact that the pressure grid is non-uniform, and therefore translation along the pressure axis is not an invariant.

Our understanding is that translational invariance is a *consequence* of the architecture of CNNs. This is because of the architecture of convolutional layers which carry out the same computation along the entire vector. Then, in the later fully connected layers, features extracted from convolutional layers are aggregated with respect to their location in the input vector using different weights. Finally, the prediction of radiation is produced.

Translation invariance is a useful property to have when the aim is to identify features regardless of their location. However, our focus was on the ability of CNNs to capture local features well rather than their ability to produce translation invariant (or equivariant) representations of the input. We believe that this is a useful feature to have given that radiative heating fields can have sharp, local features due to the presence of clouds and other radiatively active substances.

(d) `Another architecture you could try is a Resnet using skip connections.`

**Response:** We would expect that Resnet will give more accuracy in this use case. However, we expect that such a model will be even slower than the model E given that Resnet-50 has over 25 million parameters. Therefore, it is unlikely to speed up radiative transfer calculations.

(e) `You mention you use PReLUs to avoid vanishing gradients. Did you observe vanishing gradients for normal ReLUs? That would surprise me since your networks are not very deep.`

**Response:** The reviewer is correct that we did not observe vanishing gradients in all models. We only observed an indication of vanishing gradients
in our deepest model. In addition, PReLUs facilitate faster and more stable convergence. To make the comparison fair, we have chosen the same activation function in all models.

(f) `Did you observe overfitting (train MSE < valid MSE)?`
`Your validation dataset is from 2015-2016 while your`
`train dataset is from 1979-1985, which means that the`
`base state is probably changing.  This will make it`
`hard to check for overfitting since even with a model`
`that does not overfit one would expect the train loss`
`to be lower.  A better test of overfitting would be`
`to take a random subset of the train dataset.  The`
`reason I am interested in this is that I am surprised`
`a bigger network (B vs A) did not give you a better`
`score.  If you are not yet overfitting, bigger networks`
`should always result in better losses, or am I missing`
`something?  You mention overfitting when talking about`
`the generalization experiments but that still doesn't`
`explain why the skill for Dataset 1 wouldn't be better`
`for model B.`

**Response:** We agree with the reviewer that it is harder to explicitly check for over-fitting using different periods of data using standard definitions. However, we believe our approach is a more practical check of the models' ability to generalize since we would like the model to perform well in different climate states.
**2 Specific Comments**

- Line 56:  You could also mention this recent paper: https://arxiv.org/abs/2001.03151.  Also this follow up paper highlights some interesting issues:  https://onlinelibrary.wiley.com/doi/abs/10.1029/2019MS001711

  **Response:** We thank the reviewer for these references. They have been added to our bibliography.

  **Lines Changed:** 55–57

- Line 90:  Typo "columm"

  **Response:** Corrected.

  **Lines Changed:** 91

- Line 102:  I think it would be useful at this point to mention the horizontal and vertical resolution of ERA.

  **Response:** We have added the horizontal and vertical resolution.

  **Lines Changed:** 101–103

- Line 105:  So the size of your training dataset is around 12 million?  Please mention this explicitly?

  **Response:** We thank the reviewer for pointing this out. Yes, our training dataset is 12 million samples. We have added number in the manuscript.

  **Lines Changed:** 109

- Line 110:  If I understand correctly there is no vertical correlation in your perturbation, right?  Are the resulting profiles therefore "unrealistic"?  How do you think this

affects the network's generalization? Do you think
that your results are representative of the sorts of
perturbations you would experience in real climate change
(increased T)?

**Response:** We agree that our perturbed dataset is an extreme case in that it
assumes zero correlation between perturbations at different levels. However, as
mentioned previously, changed in the optical properties of the atmosphere can
be fairly "noisy" with low correlation in the vertical. In that sense, we believe our
perturbed dataset is an extreme case of a regularly occurring phenomenon. Our
choice of zero correlation aims at providing a conservative estimate: if the neural
networks can perform adequately in these conditions, they will likely perform as
well or better when there perturbations are correlated and have fewer effective
degrees of freedom. Nonetheless, we agree that these perturbed profiles affect
the network's ability to generalize, and produces issues that we have noted (Line
328 in the original manuscript). In a climate change scenario, given that our
ability to simulate the vertical and horizontal distribution of clouds is still poor, a
milder version of perturbations we have used might be relevant.

- Line 150: Actually, convolutions are a form of
  regularization by introducing an architectural constraint.
  In fact, a convolution layer is simply a subset of a fully
  connected layer with shared weights (https://medium.com/
  impactai/cnns-from-different-viewpoints-fab7f52d159c).
  But yeah, in effect you end up focusing on local features.

**Response:** We thank the reviewer for this insight. It is certainly will help us going
forward to think of CNNs in this manner.

- Table 1: I think it would be helpful to list all the
  Models

```
(including F) in the table.
```

**Response:** Model F is now in the table.

**Lines Changed:** Table 1, row 6

- ```
  Data:  Great that you included an example file in the
  repo.  But maybe you could also add instructions on how
  to download the full dataset you used.
  ```
  **Response:** We have added the URL for the ERA Interim data in the Code Availability section.

- ```
  Code:  Kudos for including code.  I looked at the
  implementation of the networks.  Maybe just a
  recommendation:  I think the code could have been a lot
  easier and clearer if you used Keras instead of plain TF.
  Especially with TF 2 its now very easy to take all the
  precoded parts of Keras and implement your own layers,
  etc.  using TF. All the manually coded layers in your code
  (matmul, etc.)  make it very hard to follow when a simple
  Dense() would suffice.  This would make the code easier to
  read for others.
  ```
  **Response:** The skeleton of keras code illustrating the NN structure for easier reading is added in the repository. It is in the model.py file line 243-255. But it is recommended to use the original implementation since it is the base for the results presented in this paper.

---

## Author Comment (AC2) · 14 Apr 2020

article [utf8]inputenc color url

[Figure]

**GMDD**

**Reply to the Reviewer 2 comments for: RadNet: Exploring deep learning architectures for longwave radiative transfer**

We thank the reviewer for his/her comments.

1. The paper explores the use of neural networks architectures
   to model radioactive transfer in global circulation models.
   The paper is interesting and well written. Authors carried
   out thorough experiments and the conclusions about the
   use of NN in the field are interesting. Additionally, the
   source code for the implementation is publicly available,
   which is really useful. I have a few concerns about the
   experiments and how the performance is calculate.

   **Response:** We thank the reviewer for their inputs and comments. We address
   the concerns below.

2. Speedups of table 2 are obtained by comparing the
   computational time of RRTMG and NN. I suppose that the
   baseline time for RRTMG is obtained in the Xeon CPU, and
   all the values are computed using this baseline. The

objective of the paper is to evaluate the advantage of
replace a RRTMG for a NN. To do that, a comparison in term
of execution time is done. But, to be fair, the comparison
of the NN should have been done with a RRTMG implementation
for GPU. In the conclusions, authors say that "a complete
rewrite of RRTMG for GPUs gives very similar performance
gains as what we have achieved in our setup using NNs
(Price et al., 2014; Mielikainen et al., 2016)". You
should include evidences to support this affirmation. For
example, in Mielikainen et al. author state that their
implementation provides a speedup of 202x on a Tesla K40
GPU, compared a single-threaded Fortran counterpart running
on Intel Xeon E5-2603 CPU. A higher speedup (370x) is
indicated in Table 2, for NN model A and a batch size of
4096 running in GTX 1080. Both GPUs architectures are
very different, so result are not directly comparable. In
fact, single precision floating point (float32) performance
is higher in the GTX 1080 that in the Tesla K40 (it's the
opposite for double precision).

**Response:** We thank the reviewer for pointing this out. We agree that our comparison does not take into consideration differences in GPU architecture. We do not have access to a GPU implementation of RRTMG, which does not allow us to make a precise estimate of the kind pointed out by the reviewer.

The main point we were trying to make is that GPUs play a significant role in accelerating RadNet. We have qualified our conclusions to highlight this fact.

**Lines Changed:** 347–349

3. Related with the previous question, Intel Xeon CPU E3-1230
   v5 is a quad-core pro- cessor (8 virtual cores). Is the

execution of RRTMG single or multi-threaded? This should
be indicated.

**Response:** The execution of RRTMG for the purposes of performance evaluation
were single-threaded. We mentioned this in the text now.

**Lines Changed:** 244

4. More details should be included about how the time is
   measured. cProfile is used for calculate the execution
   time, but with times of only 0.31 ms, as indicated in table
   2, the accuracy of the measure is not guaranteed, as a
   typical time granularity on Unix is 1 ms. Alternatives,
   as the time Python module, could provide more accurate
   measurements. Additionally, information about how many
   measurements have been done and how the final values have
   been computed should be included.

   **Response:** We thank the reviewer for pointing out this omission. The execu-
   tion time results are averaged from 10 measurements with execution of 100 000
   predictions per measurement. The above text is added to the paper.

   **Lines Changed:** 245–247

5. Introduction: To include a specific section with the
   related work, instead of including it in the introduction,
   could be beneficial

   **Response:**

6. A recent work describing other GPU-based accelerating
   methods for the RRTMG_LW could be analyzed https://www.
   mdpi.com/2076-3417/10/2/649

**Response:** We thank the reviewer for this reference. While they don't seem to achieve the same performance as previous GPU approaches, it is interesting nevertheless.

**Lines Changed:** 263, 346

7. Section 2.2. Is it realistic the perturbed dataset as has been created? Can be these values representative of a real situation?

    **Response:** It is common to have random perturbations in the optical depth due to the presence of clouds, aerosols or horizontal transport of water vapour. In some sense, our perturbed dataset tries to capture this variability. However, we agree that random perturbations at all levels is an extreme case, and it can be taken as a very strong test on the ability of our models to generalize to new situations.

8. Section 4.4. In the single column model simulation, why the NN model F is used? Why not use model A or model C?

    **Response:** The single column model uses a pressure level grid, which means that the pressure levels do not change during the course of the simulation. For this reason we use model F which interpolates the pressure levels onto a fixed pressure grid.

9. The developed code used TensorFlow, this should be indicated in the paper along with the version employed. It could be also interesting to describe the benefits of using this library in comparison with others, eg PyTorch.

    **Response:** Yes, the implementation can be done in other mainstream machine learning library such as PyTorch, Keras, etc. They could also introduce slightly different speedups. We chose TensorFlow because it is well-supported by Google and it is a mainstream machine learning platforms, especially when we started

this project. The purpose of this paper is to demonstrate that NN is capable of predicting radiative transfers with significant speedups. I believe this conclusion is still true when implementing the code in other libraries.

We have added the version employed in the manuscript.

**Lines Changed:** 172–173.

---

## Referee Report (RR1)

**Review of "RadNet 1.0: Exploring deep learning architectures for longwave radiative transfer"**

I thank the authors for their replies. However, I still have some comments and open questions which are listed below. Quotes "..." are from your reply. In particular, I still believe that the current architecture choices are not necessarily the most informative. I strongly urge the authors to actually run some new experiments as suggested below, particularly since the comparison of network architectures is probably the most interesting feature of the paper.

Network design: *I only realized from looking at your code that you are using a max-Pooling layer between every convolutional layer. Searching for "pooling" in the text only gives one result: "CNNs usually consist of three types of layers, convolutional layers, pooling layers and fully-connected layers." I think the max pooling layers should be explicitly mentioned in the text, preferably in table 1. Also, there are many CNNs without max pooling, so I think your statement is not quite accurate. See also comment below.*

"There is nothing in our setup which limits our simulations to a single column. This is because radiative transfer is assumed to have no contribution from adjacent columns (as we have described in Lines 91–92.). Our motivation to use a single column model for validation has been described in the manuscript in line 350: it allows us to study the evolution and possible errors in greater detail.
"

*Theoretically you could use your RadNet in a 3D CLiMT simulation. But as far as I can tell, it would be very difficult to create a CliMT simulation that was realistic enough so that the profiles are close to the ERA profiles. If you just plugged your ERA-trained RadNet into a simple CliMT simulation, I would expect your inputs to be very much outside of your training range. For this reason, I think training on CliMT data and then running an online 3D CliMT simulation would be the easier approach to test a 3D online run.*

"The channel number is correct. The increase is caused by the zero-padding. [...] Model E: $3*3*128+3*3*128*256+58*2*256*512+512*60 = 15531136$"

*Why is the number of channels increased for zero padding? Usually, only the non-channel dimensions are padded.*

*Also, I am not sure I understand how you compute the number of parameters for the first layer. As far as I understand your input shape is (batch_size, 62, 6). As you do in the following layers, the input channels should also be included in the computation of the parameter number, so 3*3*128*6, right?*

*Then I am confused why you have 58 in $58*2*256*512$. I thought you used max-pooling, shouldn't the size have been reduced by a factor of 4?*

"We do not think that adding padding to all conv layers will act the same as fully connected layers."

*That is not what I intended to say. In fact it wouldn't act as a fully connected layer but I still think it would be an interesting experiment because you would have a lot fewer parameters. As mentioned above, I did not realize that you used max pooling in-between every convolutional layer. I was thinking of a network like this (potentially with many more layers):*

```
model = Sequential()
model.add(Conv2D(c0_size, kernel_size=(6, 3), activation='relu', input_shape=(60,3,1), padding='same'))
model.add(Conv2D(c1_size, kernel_size=3, activation='relu', padding='same'))
model.add(Conv2D(c2_size, kernel_size=3, activation='relu', padding='same'))
model.add(Conv2D(1, kernel_size=3, activation=linear, padding='same'))
```

*That way you would avoid that huge fully connected layer after the Flatten() which I assume is the main reason the CNNs are so slow. Other ways to avoid this would be to have a much smaller number of filters in the last layer, or have more conv-max-pool layers to further decrease the size of the signal. (For an input size of 60, using 5 conv layers would decrease the signal size to around 2. Flattening then would result in a much smaller vector.) As it is currently your experiments do not support your conclusions that CNNs cause a performance loss because I assume that the performance loss comes from the big fully-connected layer rather than the conv layers.*

"We would expect that Resnet will give more accuracy
in this use case. However, we expect that such a model will be even
slower than the model E given that Resnet-50 has over 25 million
Parameters."

*Resnet does not specifically refer to the Resnet-50 used for image classification. Rather it just refers to the idea of using skip connections which come at basically no extra cost except for one addition. I would suggest trying out a network that looks something like this (here I am using the Keras functional API):*

```
x = inp = Input(shape=(60, 3,))
x = Conv2d(size, kernel_size, activation='relu')(x)
x = Conv2d(size, kernel_size, activation='relu')(x)
x = Conv2d(size, kernel_size, activation='relu')(x)
x = Add()([inp, x])
outp = Conv2d(1, kernel_size, activation='relu')(x)
```

*You could also add several resblocks (something like 5 not 50). You could either do this in a fully convolutional way as in my code, or add pooling layers between the resblocks*

*(bottlenecks) and use a fully connected layer in the end. In general, I would strongly suggest that you at least test the experiments I suggested since it shouldn't be much extra work.*

*Also, you did not directly reply to my comment about the scores of model A and B: " I am surprised a bigger network (B vs A) did not give you a better score. If you are not yet overfitting, bigger networks should always result in better losses, or am I missing something? You mention overfitting when talking about the generalization experiments but that still doesn't explain why the skill for Dataset 1 wouldn't be better for model B." Do you have any explanation why model B isn't better than model A? Is the training loss lower for B than for A?*

---

## Referee Report (RR2)

Again, I thank the authors for addressing my comments. I have to admit I am disappointed that, despite asking twice, the authors refused to perform any new experiments, even though this should only take a few hours. I still believe that adding more network experiments would have made the paper much more interesting. I still have one complaint about the CNN experiments and their interpretation (see below). However, the paper probably has just enough merit to be published as is. I leave it to the authors to decide what to do with my final comments.

CNN Experiments: I only just realized 2D convolutions were used. I guess I just assumed that the convolutions are 1D. This choice is odd. From my point of view (and that of others I believe) the more natural choice would have been to use 1D convolutions and treat the different variables as channels. Again, a comparison of these two would have been really interesting, but I assume that the authors aren't willing to perform new experiments… In any case, one of the key conclusions of the paper (it's in the abstract) is that CNNs are more accurate but slow. But what I think makes the CNN models slow is the huge Dense layer, I would assume. Performing a fully-convolutional experiment would have shed light on this. In the absence of these clarifying experiments I would suggest being more careful with the wording because, from my point of view, the experiments do not support the current claim.

---

## Author Response (AR2)

**Reply to the Reviewer 1 comments for: RadNet: Exploring deep learning architectures for longwave radiative transfer**

We thank the reviewer for his/her comments.

**1 Comments**

1. I thank the authors for their replies.  However, I still have
   some comments and open questions which are listed below.  Quotes
   "..." are from your reply.  In particular, I still believe that
   the current architecture choices are not necessarily the most
   informative.  I strongly urge the authors to actually run some
   new experiments as suggested below, particularly since the comparison
   of network architectures is probably the most interesting feature
   of the paper.

   **Response:** We thank the reviewer for their effort to understand our work in depth and provide constructive ways in which our work could be improved.

   The core messages of the paper was twofold: 1) To explore CNN and MLP architectures (in terms of their performance and ability to generalize) and 2) To understand the physical impact of NN errors on simulation. We believe our physically-inspired approach to evaluate generalizability of NNs using a perturbed dataset is also a contribution of this paper. As we have emphasized in the paper, the fluid dynamical core present in a GCM "redistributes" errors and makes it difficult to understand the direct impact of NN errors within a column. We believe that performing single column experiments is a novel approach to testing NNs developed for climate modelling, and we hope our work will encourage others to adopt a hierarchical approach to testing models (A time-tested approach within climate science for many decades).

   Furthermore, we have shown in our modelling experiments that traditional notions of accuracy (such as MSE) which are used to evaluate NNs do not necessarily translate to accurate simulations. We hope that the paper is evaluated based on all its contributions, rather than only the comparison of architectures.

We agree with the reviewer that there could be other architectural options which could improve the performance over the simple CNN architectures we have used. For instance, based on the recent improvements on ImageNet models, we see an improvement on the accuracy of 10 - 20 percent and a reduction of the number of parameters 5 to 10 times in recent years. However, since our task is to some extent different from an image recognition task, the best approach is unknown and further investigation of different models and techniques need to be explored in the context of using NNs for climate modelling.

2. `Network design:  I only realized from looking at your code that you are using a max-Pooling layer between every convolutional layer.  Searching for "pooling" in the text only gives one result: "CNNs usually consist of three types of layers, convolutional layers, pooling layers and fully-connected layers." I think the max pooling layers should be explicitly mentioned in the text, preferably in table 1.  Also, there are many CNNs without max pooling, so I think your statement is not quite accurate.  See also comment below.`

   **Response:** We thank the reviewer for taking their time to go through our code.

   We have not used max-pooling for models used in the paper. However, we understand how the skeleton code that we have provided can give rise to this misunderstanding. We apologize for the misunderstanding. As we have stated in the README file: "The model is defined in radiation/model.py. Since in the paper, we have a lot of configurations of the models with varying number of convolutional layers and the input size. Thus, it is up to the user to modify the model structure to suits his/her needs by just commenting/uncomment the code or modifying a couple lines of code." We note that we tried other experiments as well, such as batch normalizations. These experiments were not included in the final version of the paper to ensure we deliver clear messages about the performance of NNs in climate modelling, and methods to validate accuracy and fidelity.

   We have updated the example code (line 387) to add comments to ensure such a misunderstanding does not occur when others go through our code in the future. And we have updated the paper in line 170 to explicitly mention that we have not used pooling layers.

3. `"There is nothing in our setup which limits our simulations to a single column.  This is because radiative transfer is assumed to have no contribution from adjacent columns (as we have described in Lines 91{92.).  Our motivation to use a single column model for validation has been described in the manuscript in line 350: it allows us to study the evolution and possible errors in greater detail." Theoretically you could use your RadNet in a 3D CLiMT`

simulation. But as far as I can tell, it would be very difficult
to create a CliMT simulation that was realistic enough so that
the profiles are close to the ERA profiles. If you just plugged
your ERA-trained RadNet into a simple CliMT simulation, I would
expect your inputs to be very much outside of your training range.
For this reason, I think training on CliMT data and then running
an online 3D CliMT simulation would be the easier approach to
test a 3D online run.

**Response:** We agree with the concerns raised by the reviewer. Our
previous response was not intended to suggest that the NNs that we have
trained currently will be transparently work in a GCM setting. We meant
to say that our training methodology, which focuses on training a network
on data from single columns, should work even when training a model for a
GCM, since columns do not interact radiatively within a GCM simulation.

We agree that using CliMT simulation data for RadNet might be a more
viable strategy, and we intend to pursue this question in future research.

4. "The channel number is correct. The increase is caused by the
   zero-padding. [...] Model E: 3x3x128+3x3x128x256+58x2x256x512+512x60
   = 15531136" Why is the number of channels increased for zero padding?
   Usually, only the non-channel dimensions are padded.

   **Response:** We think there may be a misunderstanding of channels here.
   As from your first comment, "Model E is the same as Model C but with
   zero-padding but why do the number of channels also increase in the table
   from 4 to 6?". Perhaps you refer to the second dimension of the input to
   be the channel. By this interpretation, we mean that the first and second
   dimensions are increased by 0 paddings from 60x4 to 62x6.

5. Also, I am not sure I understand how you compute the number of
   parameters for the first layer. As far as I understand your input
   shape is (batch_size, 62, 6) As you do in the following layers,
   the input channels should also be included in the computation
   of the parameter number, so 3x3x128x6, right?

   **Response:** We do not think so. Let us try to explain it like this. The
   input shape is (batch_size, 62, 6, 1) if you make it comparable with later
   layers. So that the weights are 3x3x1x128 (given 128 conv filters). After
   the first conv layer, the input for the next layer becomes (batch_size, 61,
   5, 128). Thus, the weights for the next layer are 3x3x128x256 (given 256
   conv filters).

6. Then I am confused why you have 58 in 58x2x256x512. I thought
   you used max-pooling, should not the size have been reduced by
   a factor of 4?

   **Response:** As mentioned previously, we do not use max-pooling in our
   models. Therefore, the size is not reduced.

7. "We do not think that adding padding to all conv layers will act the same as fully connected layers." That is not what I intended to say. In fact it would not act as a fully connected layer but I still think it would be an interesting experiment because you would have a lot fewer parameters. As mentioned above, I did not realize that you used max pooling in-between every convolutional layer. I was thinking of a network like this (potentially with many more layers): CODE BLOCK. That way you would avoid that huge fully connected layer after the Flatten() which I assume is the main reason the CNNs are so slow. Other ways to avoid this would be to have a much smaller number of filters in the last layer, or have more conv-max-pool layers to further decrease the size of the signal. (For an input size of 60, using 5 conv layers would decrease the signal size to around 2. Flattening then would result in a much smaller vector.) As it is currently your experiments do not support your conclusions that CNNs cause a performance loss because I assume that the performance loss comes from the big fully-connected layer rather than the conv layers.

   **Response:**

   It is true that adding pooling or using a larger stride can significantly reduce the signal size for the later layers. And it will make the model undoubtedly faster. But for the accuracy, we do not know before testing. Thus, we consider this as another approach to evaluate the tradeoff between accuracy and speed. And, it is definitely something one should try to investigate an optimal architecture for this task.

8. "We would expect that Resnet will give more accuracy in this use case. However, we expect that such a model will be even slower than the model E given that Resnet-50 has over 25 million Parameters." Resnet does not specifically refer to the Resnet-50 used for image classification. Rather it just refers to the idea of using skip connections which come at basically no extra cost except for one addition. I would suggest trying out a network that looks something like this (here I am using the Keras functional API): CODE BLOCK. You could also add several resblocks (something like 5 not 50). You could either do this in a fully convolutional way as in my code, or add pooling layers between the resblocks (bottlenecks) and use a fully connected layer in the end. In general, I would strongly suggest that you at least test the experiments I suggested since it should not be much extra work.

   **Response:** We agree that using Residual blocks could be a potential direction to further investigate the capability of the model in terms of accuracy and speed. We understand that Residual blocks (2015) are a newer idea than simple CNN. We agree there is a lot of work left to push

the boundary of finding the best network. However, we need to have a definition on the requirements of speedups and accuracy before approaching the solution. For example, we might want to aim at 100x speedup while finding the best accuracy. But these requirements are defined case by case. Thus, we would like to keep the paper to deliver the simplest idea, and leave the question of finding the best model for future work. We have incorporated the reviewer's suggestions in the discussion section as a pointer for future work. We thank the reviewer for these suggestions – they enrich the paper and provide the reader a clear way forward in improving upon our results.

9. Also, you did not directly reply to my comment about the scores of model A and B: " I am surprised a bigger network (B vs A) did not give you a better score. If you are not yet overfitting, bigger networks should always result in better losses, or am I missing something? You mention overfitting when talking about the generalization experiments but that still doesn't explain why the skill for Dataset 1 wouldn't be better for model B." Do you have any explanation why model B isn't better than model A? Is the training loss lower for B than for A?

   **Response:** We believe that over-fitting is the cause of this result. We have extended the explanation regarding this issue in the paper (Line 228-232). Specifically, the strongest indication is the results in Figure 4, which has shown that model B has significantly higher error than model A. Given that Dataset 2 is more perturbed than Dataset 1 and more parameters in model B makes it easier to fit the training data (Dataset 1), a larger error is expected while evaluating against Dataset 2.

   We do not have a confirmed answer for why model B performs slightly worse than model A on the same Dataset 1 or Dataset 2. We would guess it is because of the over-fitting issue indicated in Figure 4. Yes, the training loss is slightly lower for model B than model A.

[revised manuscript text omitted]

---

## Author Response (AR3)

**Reply to Reviewer 1 for the manuscript**
**RadNet: Exploring deep learning architectures for longwave radiative transfer**

We thank the reviewer for their suggestions and their careful evaluation of our work. We agree with the reviewer that exploring different architecture optimizations would be interesting; however, within the framework we have proposed, evaluating a model is not just a matter of a few hours of work. It involves not only evaluating the offline RMS error and performance gain of any such optimization, but also evaluating its performance with a single column model and ensuring the model works well with other components to produce stable integration. In this light, we thank the reviewer for their understanding.

**1  Comments**

1. Again, I thank the authors for addressing my comments.  I have
   to admit I am disappointed that, despite asking twice, the authors
   refused to perform any new experiments, even though this should
   only take a few hours.  I still believe that adding more network
   experiments would have made the paper much more interesting.  I
   still have one complaint about the CNN experiments and their interpretation
   (see below).  However, the paper probably has just enough merit
   to be published as is.  I leave it to the authors to decide what
   to do with my final comments.
   CNN Experiments:  I only just realized 2D convolutions were used.
   I guess I just assumed that the convolutions are 1D. This choice
   is odd.  From my point of view (and that of others I believe)
   the more natural choice would have been to use 1D convolutions
   and treat the different variables as channels.  Again, a comparison
   of these two would have been really interesting, but I assume
   that the authors aren't willing to perform new experiments...

   **Response:**   We thank the reviewer for taking their thorough under-
   standing of our work and the effort they have made to suggest ways to
   optimise our experiments, especially the CNN-based ones. Our motiva-
   tion to use 2D filters instead of 1D was to give more freedom to the CNN

to learn the relations among the variables at the same vertical level. 2D filters allow the CNN to learn, for instance, that co-located local changes in temperature and specific humidity (in the presence of an inversion, for example) produce a different solution as compared to changes that are not co-located.

As the reviewer points out, this choice of permitting the CNN to learn the effects of correlations between different fields leads to slower performance. A detailed study of the accuracy-performance trade-off for such a choice would indeed be interesting to explore.

We have now explicitly pointed out this design choice in our paper (Line 166–168) and highlight the tradeoff it represents in the discussion section (Line 341–344).

2.  In any case, one of the key conclusions of the paper (it's in
    the abstract) is that CNNs are more accurate but slow.  But what
    I think makes the CNN models slow is the huge Dense layer, I would
    assume.  Performing a fully-convolutional experiment would have
    shed light on this.  In the absence of these clarifying experiments
    I would suggest being more careful with the wording because, from
    my point of view, the experiments do not support the current claim.

**Response:**    We agree with the reviewer.  We have pointed out the limitations of our conclusions in line 362–366. We have also noted in the abstract that our conclusions hold only for conventional CNNs.

[revised manuscript text omitted]